# Development of a Protocol for Biomechanical Gait Analysis in Asian Elephants Using the Triaxial Inertial Measurement Unit (IMU)

**DOI:** 10.3390/vetsci9080432

**Published:** 2022-08-15

**Authors:** Kittichai Wantanajittikul, Chatchote Thitaram, Siripat Khammesri, Siriphan Kongsawasdi

**Affiliations:** 1Department of Radiologic Technology, Faculty of Associated Medical Sciences, Chiang Mai University, Chiang Mai 50200, Thailand; 2Center of Elephant and Wildlife Health, Chiang Mai University, Chiang Mai 50200, Thailand; 3Department of Companion Animals and Wildlife Clinics, Faculty of Veterinary Medicine, Chiang Mai University, Chiang Mai 50100, Thailand; 4Department of Physical Therapy, Faculty of Associated Medical Sciences, Chiang Mai University, Chiang Mai 50200, Thailand

**Keywords:** biomechanics, elephant, gait, IMU

## Abstract

**Simple Summary:**

In general practice, the veterinarian and caregiver usually detect lameness in elephants from observation of any discomforting characteristics when walking. Currently, motion analysis can offer an objective method to evaluate normal and abnormal gait accurately, thus identifying changes in some characteristics when walking. This report aimed to introduce a recent technology utilizing wireless sensors for quantitative analysis of joint angles during the gait cycle in Asian elephants. To enable three-dimensional limb segment motion, a triaxial inertial measurement unit (IMU) is equipped with three sensor types: an accelerometer, a gyroscope, and a magnetometer. Therefore, we hope that this portable sensor-based system can help clinicians in diagnosis, especially in the early stages of lameness. Moreover, with wireless signal transmission, the system is clinically applicable for use in all areas where electricity is available.

**Abstract:**

Gait analysis is a method of gathering quantitative information to assist in determining the cause of abnormal gait for the purpose of making treatment decisions in veterinary medicine. Recent technology has offered the wearable wireless sensor of an inertial measurement unit (IMU) for determining gait parameters. This study proposed the use of a triaxial IMU, comprising an accelerometer, a gyroscope, and a magnetometer, for detecting three-dimensional limb segment motion (*XYZ* axis) during the gait cycle in Asian elephants. A new algorithm was developed to estimate the kinematic parameter that represents each limb segment of the forelimbs and hindlimbs while walking at a comfortable speed. For future use, this study aimed to create a new prototype of the IMU with a configuration that is tailored to the elephant and apply machine learning in an effort to achieve greater precision.

## 1. Introduction

Elephants have a unique locomotion pattern aimed to stabilize their huge body mass and conserve metabolic energy [1,2]. Understanding the body structure and function of the locomotory system has enabled a mechanism of how the elephant walks which may be applied for clinical purposes; for example, they have pillar-like legs that allow them to support their enormous weight [3]. This knowledge may be applied for clinical purposes. The kinematics and kinetics of an elephant gait is of interest for understanding how the limb segments and joints move and stabilize in specific ways, despite their gigantic mass and need to conserve energy when walking. The study of kinetics focuses on the forces applied to the body, acceleration, and energy that influence how an individual moves. It is typically necessary to conduct studies in a motion laboratory with a force plate system in order to calibrate the ground reaction force and relate parameters that interact with the muscles, ligaments, tendons, and bony structures [4,5,6]. Kinematics of gait involves the joint angles, accompanying linear acceleration, angular velocities, and the position of segments through each phase of the gait [7]. Both kinetics and kinematics underline mechanisms that minimize muscle work and the metabolic cost of movement, which also relate to neural control strategies [4,5,6].

The systematic study of biomechanical gait analyzes dynamic posture and coordination while moving. In clinical practice, clinicians utilize it to evaluate how the body moves in order to identify any alterations that cause abnormal gait [7]. A number of studies have documented biomechanical variables of elephant locomotion such as limb movement, joint angles, angular displacement, and ground reaction force. Hildebrand and Hurley initiated the study of quantitative locomotion and gait analysis in elephants in 1985. By using camera captures, they created a model to calibrate the kinetic energy of the legs when walking at near maximal speed [8]. Hutchinson and Ren et al. presented numerous studies, particularly to explain the kinematics and kinetics of limb mechanics that change at various speeds in order to conserve the energy cost in locomotion [1,9,10,11]. The quantitative gait analysis currently includes computerized video cameras, infrared markers, and force platforms that provide objective information to support clinicians in diagnosing and monitoring therapeutic intervention. Gait analysis has been conducted in a laboratory setting for humans and certain species of animals such as horses [12,13] and dogs [6,14]. However, conducting experiments on larger animals such as elephants is not feasible in a laboratory setting due to their size and the costs. Veterinarians, therefore, observed elephant gait in clinical practice and only gathered subjective information from one expert. Since the start of the 20th century, a novel technology of wearable sensors has been utilized to monitor locomotion and mobility in daily life. The microelectromechanical system (MEMS) and inertial measurement unit (IMU) have become efficient tools for biomechanical evaluation of gait kinematics and other types of locomotion [15,16,17]. The system consists of an accelerometer, a gyroscope, and a magnetometer, and these components are arranged in a triaxial configuration to provide measurements along three axes for a total of nine degrees of freedom. A triaxial accelerometer measures the linear acceleration of movements in three dimensions from either body motion or gravity. A gyroscope measures the angular velocity by yielding an estimation of sensor orientation at each point in time, which is relative to the local frame of its original orientation, and measures the rate of turnaround of three orthogonal axes: roll (X), pitch (Y), and yaw (Z). A magnetometer is a device for measuring the amplitude and direction of the local magnetic field in three dimensions. Since the magnetometer measures yaw angle rotation, it can be calibrated to gyroscope data in order to reduce large drift and provide appropriate calculations of dynamic orientation [16,17]. As a result, the integrated signal from the IMU represents the kinematic parameters of gait, which focuses on position and orientation of body segments [13]. In elephants, Ren and Hutchinson were the first group to develop the inertial sensor that integrated with three-dimensional accelerometers and gyroscopes to measure kinetic and kinematic (angular velocity and acceleration) parameters during normal (1.3 m per second) to moderate speed (3.07 m per second), and quantify how they modulated the gait pattern and mechanical energy across the alteration of gait speed [11].

Due to limited studies using this technology to evaluate elephant gait, this study aimed to introduce the development of the protocol and algorithm to determine the kinematic characteristics of an Asian elephant’s gait by using wearable inertial sensors. This report is an extended version of the authors’ work. It presents a method for objectively identifying gait alteration, which corresponds to a particular pathology that causes lameness; thus, improving decisions of the veterinarian when diagnosing and treating.

## 2. Materials and Methods

### 2.1. Study Animals

This study analyzed two elephants from the National Elephant Institute, Forest Industry Organization, Thailand. They were given a physical examination by a skilled veterinarian. A 64-year-old female elephant with a body weight (BW) of 2855 kg and body condition score (BCS) of 4 had no lameness in any limbs, whereas a 4-year-old female elephant (BW 970 kg, BCS 3.5) suffered a left carpal amputation due to injury. This study was approved by the Animal Care and Use Committee, Faculty of Veterinary Medicine, Chiang Mai University (Research ID R/20; July 2019).

### 2.2. Data Collection and Analysis

Data were processed when the elephants walked in a particular straight line for 20 m on flat concrete at a consistent speed and usual pace, guided by a mahout without physical contact Figure 1b. Practice trials were performed to ensure that the elephants were familiar with the experimental setting and could walk at a pace that was preferable for them. Eight IMU sensors (STT Ingeniería Y Sistemas, San Sebastián, Spain) were attached to the skin; two sensors on both forelimbs along the proximal humerus and two on the radius bones. For the hindlimbs, two sensors were placed along the proximal part of the femur and two on the tibia for proximal and distal motion in Figure 1a. Sensor signals were transmitted to software via Wi-Fi. Data tracking was calculated at the midpoint for 5 consecutive walking cycles to reduce the effects of the starting acceleration and ending deceleration.

To avoid variations in data collection, it was necessary to standardize the data. As illustrated in Figure 2a, only 5 cycles of the right proximal forelimb were selected to normalize the walking duration in order to confirm that the walking speed of the elephant was steady.

*t*_1_ and *t*_2_ were chosen as the start and the end points, respectively. The duration was then converted to a range of 0 to 1 using Equation (1): (1)$${t^{\prime }}_{i}=\frac{\left(t_{i}-t_{1}\right)}{\left(t_{2}-t_{1}\right)},$$
where ${t^{\prime }}_{i}$ is the normalized *t_i_,* and *t_i_* is the time at point *i*. As indicated in Figure 2b, the *X* axis is scaled from 0 to 1.

The initial value from each IMU sensor might differ since the absolute angle is determined individually. As a result, the collected data should be calibrated as illustrated in Figure 3a.

The acquired data from each IMU sensor were calibrated by subtracting their means, as indicated in Equation (2):(2)$${A^{\prime }}_{i}=A_{i}-\left(\frac{\sum_{k=1}^{N}A_{k}}{N}\right),$$
where *A_i_* represents the absolute angle at point *i* and ${A^{\prime }}_{i}$ represents the calibrated *A_i_.* Figure 3b showed that the plots were calibrated into the same range, which made the analytical procedure easier.

## 3. Results

The elephants walked at a comfortable speed at their own normal pace while being guided by their individual mahout at an average speed of 1.24 m per second, which ranged from 1.10–1.51 m per second. Figure 4 represents the gait pattern of the 64-year-old female elephant with no pathologies involving gait, as confirmed by two experienced veterinarians. The amplitude of the proximal and distal limb segments was demonstrated for the right (solid line) and left (dash line) forelimbs and hindlimbs. Graphical data demonstrated the synchronized pattern of each stride, and the existing stance and swing phase between left and right limbs. The movement amplitude of all segments and the period of transition between stance and swing were symmetrical in corresponding time. With the present methodology of analysis, it was found that the forelimb continues to sustain some weight during the swing of the hindlimb, and may attempt to distribute weight across three legs while the body accelerates forward, which is consistent with previous studies [2,10].

Figure 5 illustrates the gait pattern of the four-year-old elephant who was trapped by a snare at the left carpal joint when she was one year old. A transmetacarpal, partial amputation was performed. As a result, a prosthetic shoe was applied as a substitute for the absent component and the elephant regained the ability to walk. The absolute angle of this case was evaluated recently from the report of a previous trial [18]. According to existing data, the amputated side on the left (dash line) moved asynchronously and asymmetrically with the right normal side (solid line), in the proximal and distal segments of either the forelimb or distal hindlimb. The left-side limb advanced quicker and with greater amplitude than the right one. It was assumed that the elephant was uncomfortable with the prosthetic limb, and consequently, moved the right forelimb faster during midstance to shift weight away from the unpleasant left side (Figure 5). Two years after the injury, she was unable to bear weight properly on her left forelimb, which necessitated a weight shift to the right side, and showed an abnormally bowed leg (Figure 6).

## 4. Discussion

In this study, the kinematic metric was represented as the graphical motion of the joint segment, based on changes in absolute rotation angle. This joint segment motion represents the virtually normal movement of humans and animals, which involves the rotation of body segments. Each gait cycle comprised two phases: stance and swing. The stance phase occupied 60% of the gait cycle, which began with a foot strike and ended with the same foot lifting-off. Over the phase, the foot made contact with the ground and the limb bore the weight. The swing phase included approximately 38% of the gait cycle, in which the foot moved forward while in the air and did not put weight on the ground [7]. This study’s findings are consistent with prior research that revealed elephants use an inverted pendulum mechanism to move and that the limb phase was relatively constant while walking [19,20]. The limb bone orientation of elephants is almost vertical and generally adopts straighter limbs with a small degree of movement range. Previous studies by Ren [9] revealed that the forelimb joints of elephants are extended highly while in the load-bearing phase of the stance. All of these mechanical advantages allow elephants to maintain stability and utilize muscular work and energy efficiently.

Due to the metric of this study, a certain limitation was apparent. The absolute angle contains components of inertial and gravitational acceleration so that the placement of the sensor may affect the vertical orientation signal. It is the plan for future studies to assign each sensor individually in order to overcome the variance of signals.

This study presented a novel method for analyzing the biomechanics of the gait of elephants by using the wireless IMU sensor. It is able to overcome the limitations of the conventional method, which requires professional experience, tends to provide subjective findings or quantitative evaluation, and requires complex and expensive laboratory settings. The absolute angle of the limb segment, which is a three-dimensional kinematic metric, can describe mammalian locomotion features that are similar to naturally integrated movement. This study proposes employing machine learning techniques in the future, in order to achieve better outcomes, for use as clinical diagnostic and assessment tools in determining the efficacy of treatment.

## Figures and Tables

**Figure 1 vetsci-09-00432-f001:**
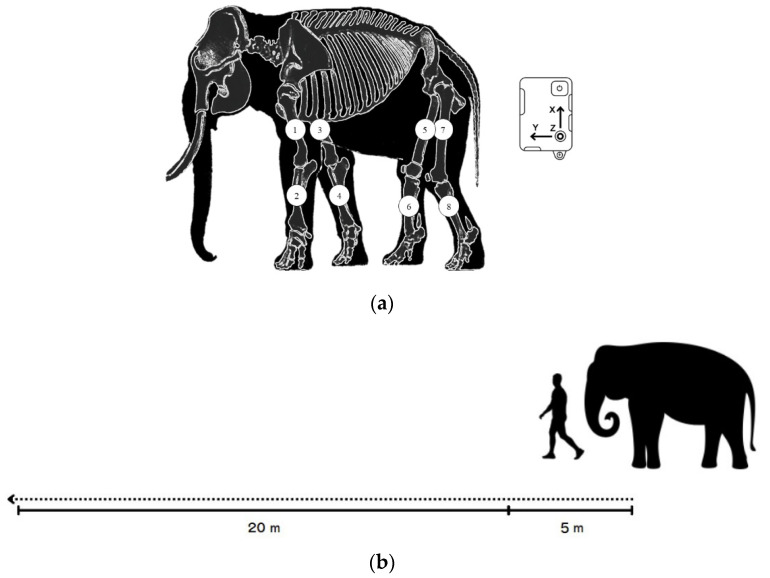
(**a**) The placement of the eight sensors on each forelimb (1,3 placed along the humerus; 2,4 on the radius) and hindlimb (5,7 along the femur and 6,8 along the tibia of the hindlimbs) and schematic of the pathway of data collection; (**b**) 5 m at the beginning to reduce the effect of an acceleration phase and 20 m for data collection.

**Figure 2 vetsci-09-00432-f002:**
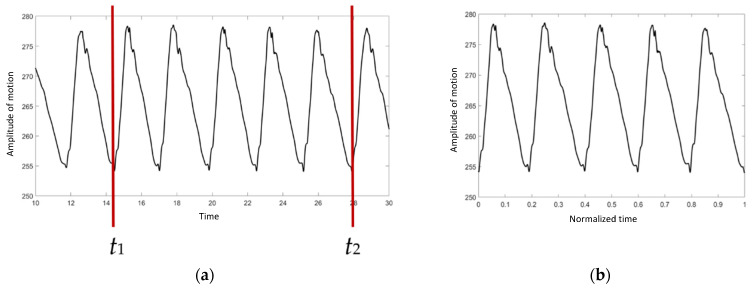
The graph of absolute angle versus time: (**a**) before the normalization procedure, (**b**) after the normalization procedure.

**Figure 3 vetsci-09-00432-f003:**
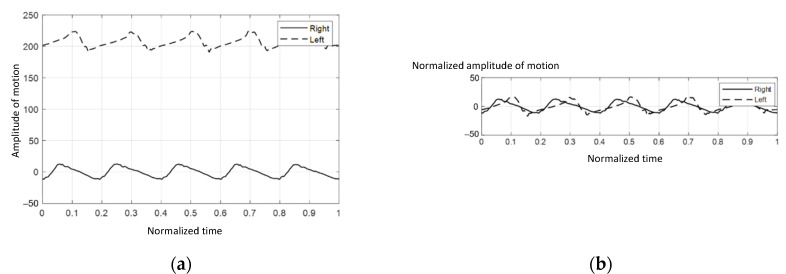
The graphs of absolute angle versus time: (**a**) before the calibration procedure, (**b**) after the calibration procedure.

**Figure 4 vetsci-09-00432-f004:**
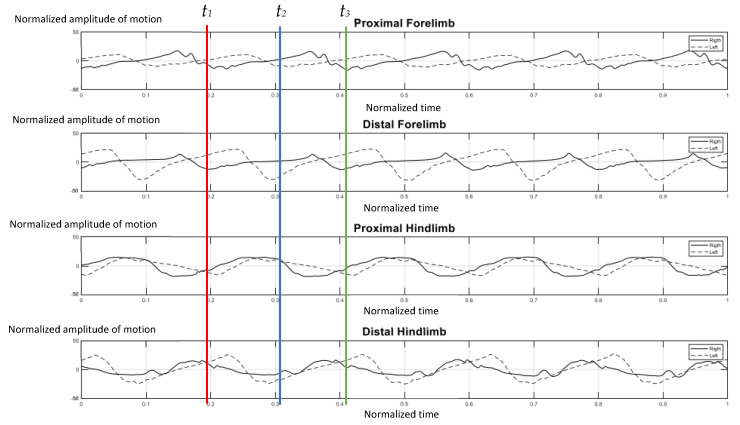
Graphical illustration of the amplitude of forelimb and hindlimb absolute rotation angle (*Y* axis) over the time-cycle of the gait (*X* axis) of the 64-year-old female non-lame elephant. The amplitude of absolute angle of proximal and distal limb segments was demonstrated for the right (solid line) and left (dash line) of forelimbs and hindlimbs. The stance phase of right forelimb represented by *t*_1_–*t*_2_ and *t*_2_–*t*_3_ was the period of the swing phase of right fore-limb.

**Figure 5 vetsci-09-00432-f005:**
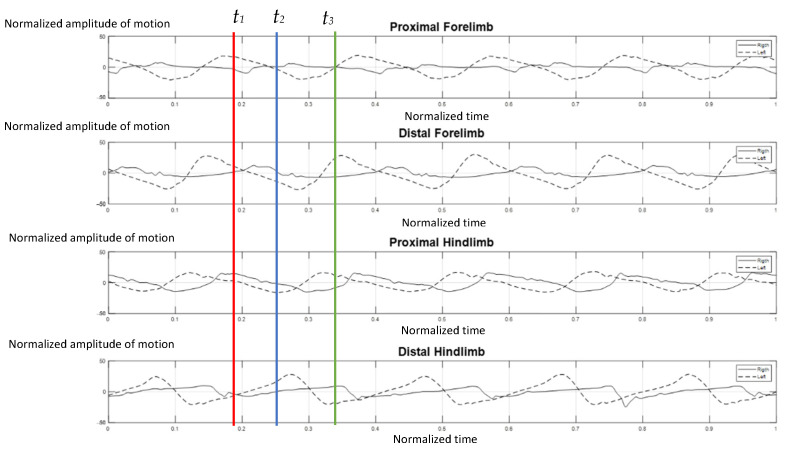
Graphical demonstration of amplitude of forelimb and hindlimb absolute rotation angle over the time-cycle of the gait of the 4-year-old female elephant with left carpal amputation. The amplitude of absolute angle of proximal and distal limb segments was demonstrated for the right (solid line) and left (dash line) of forelimbs and hindlimbs. The stance phase of right forelimb represented by *t*_1_–*t*_2_ and *t*_2_–*t*_3_ was the period of the swing phase of right fore-limb.

**Figure 6 vetsci-09-00432-f006:**
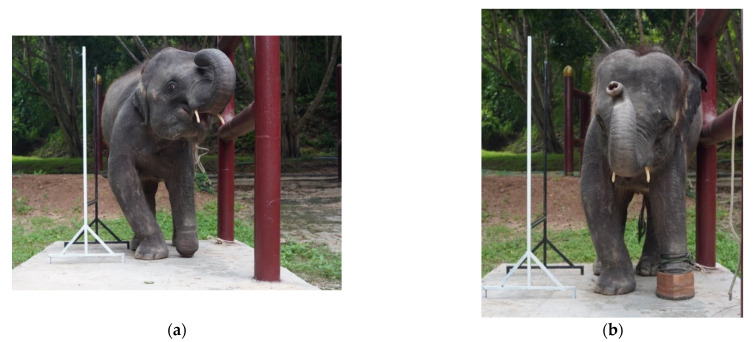
Development of a bowed right forelimb compared between standing without (**a**) and with a prosthetic shoe (**b**).

## Data Availability

The data analyzed for the study are available from the corresponding author upon reasonable request.

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
