# Peer review of "Development of a Protocol for Biomechanical Gait Analysis in Asian Elephants Using the Triaxial Inertial Measurement Unit (IMU)"

_vetsci, 2022, doi:10.3390/vetsci9080432_

Round 1

Reviewer 1 Report

 This is a pilot study, which could have utilized a few more elephants to increase the scientific soundness, but still demonstrates its feasibility for future development.  

This is an introduction of a technology which has been accepted for horses and dogs to be able to make objective evaluations of lameness.  However, it should be stated in the concluding paragraph, that this technology is useful for elephants in human care with a hands-on management system.  But would not be useful for most elephants in other styles of elephant management systems. 

Reviewer 2 Report

this paper has some serious flaws in that it ignore previous published research on elephant locomotion (see Hutchinson et al 2006 JEB 209: 3812-3827)

the sample size is very small with a n=1 limiting any major conclusions that can be drawn from the study - working with an injured animals as well limits the conclusions and inferences that can be drawn - I would suggest increasing the sample size and examining a novel question on elephant gait before considering resubmission of the paper

Specific Points:

the manuscript need editing for English and grammar

line 37 - this isn't true papers have been published on elephant gait analysis for African and asian elephants using accelerometry:

Alexander, R. McN. and Maloiy, G. M. O. (1989). Locomotion of African

mammals. Symp. Zool. Soc. Lond. 61, 163-180.

Christian, A., Müller, R. H. G., Christian, G. and Preuschoft, H. (1999).

Limb swinging in elephants and giraffes and implications for the reconstruction of limb movements and speed estimates in large dinosaurs. Mitt. Mus. Nat. Berl. Geowiss. Reihe 2, 81-90.

Hildebrand, M. and Hurley, J. P. (1985). Energy of the oscillating legs of a

fast-moving cheetah, pronghorn, jackrabbit, and elephant. J. Morphol184,

23-31.

Hutchinson, J. R., Famini, D., Lair, R. and Kram, R. (2003). Biomechanics:

are fast-moving elephants really running? Nature 422, 493-494.

line 83 - this figure adds little to the paper overall 

line 92 - the point of this figure isn't clear

line 111 - define comfortable speed

line 132 - you need to fully explain your figure legends 

line 153 - what are the novel findings of this study ? 

Reviewer 3 Report

Dear Authors

The work presented for review, seems interesting and the topic addressed is timely and necessary. Before the possible acceptance of the work, offers some comments , which in my opinion will improve the quality of this manuscript.

In the Introduction section:
1. please , the authors should explain to potential readers what the terms kinematics and kinetics mean.
2 The organ of locomotion is an extremely interesting and complex subject whose main role is mechanical function. The elementary parts that make up the organ of locomotion are bones and cartilage connected to each other by joints and reinforced by ligaments and other connections. Thus, the skeletal system is the passive part of the organ of locomotion, in order to set it in motion you need its active part - the muscular system.  So please, for the readers, introduce a supporting drawing describing at least the different segments of the limbs and indicating the different joints.

In the Materials and Methods section:

The authors did not describe in Fig. 1 , what is in its part a and what is in b. Explain what is meant by 1, 2, 3, 4, 5, 6, 7, 8?
Figs. 2, 4 and 5 are unreadable , due to poor quality and size. Please improve the quality of these Figs.

In the Results and Discussion section:

Correct in the sentence: Considering a four-year-old young elephant who was trapped by a snare at the left carpal joint when she was one year old. The trans-metacarpal, partial foot amputation was performed-not the foot but the hand!

Regards
